# Evaluation of the Multipath Environment Using Electromagnetic-Absorbing Materials at Continuous GNSS Stations

**DOI:** 10.3390/s22093384

**Published:** 2022-04-28

**Authors:** Addisu Hunegnaw, Felix Norman Teferle

**Affiliations:** Department of Engineering, Faculty of Science, Technology and Medicine, University of Luxembourg, L-1359 Luxembourg, Luxembourg; norman.teferle@uni.lu

**Keywords:** global navigation satellite system (GNSS), multipath, site-specific effects, microwave-absorbing material (MAM), GNSS reference station

## Abstract

To date, no universal modelling technique is available to mitigate the effect of site-specific multipaths in high-precision global navigation satellite system (GNSS) data processing. Multipaths affect both carrier-phase and code/pseudorange measurements, and the errors can propagate and cause position biases. This paper presents the use of an Eccosorb AN-W-79 microwave-absorbing material mounted around a GNSS antenna that reflects less than −17 dB of normal incident energy above a frequency of 600 MHz. To verify the feasibility and effectiveness of the Eccosorb, we installed two close stations by continuously operating multi-GNSS (BeiDou, GLONASS, Galileo and GPS) in a challenging location. One station is equipped with the Eccosorb AN-W-79, covering a square area of 3.35 m${}^{2}$ around the antenna, and the second station operates without it. The standard deviation reductions from single point positioning estimates are significant for all the individual GNSS solutions for the station equipped with microwave-absorbing material. The reductions are as follows: for GPS, between 15% and 23%; for Galileo, between 22% and 45%; for GLONASS, 22%; and for BeiDou, 4%. Furthermore, we assess the influence of multipaths by analysing the linear combinations of code and carrier phase measurements for various GNSS frequencies. The Galileo code multipath shows a reduction of more than 60% for the station with microwave-absorbing material. For GLONASS, particularly for the GLOM3X and GLOM1P code multipath combinations, the reduction reaches 50%, depending on the observation code types. For BeiDou, the reduction is more than 30%, and for GPS, it reaches between 20% and 40%. The Eccosorb AN-W-79 microwave-absorbing material shows convincing results in reducing the code multipath noise level. Again, using microwave-absorbing material leads to an improvement between 15% and 60% in carrier phase cycle slips. The carrier-phase multipath contents on the post-fit residuals from the processed GNSS solutions show a relative RMS reduction of 13% for Galileo and 9% for GLONASS and GPS when using the microwave-absorbing material. This study also presents power spectral contents from residual signal-to-noise ratio time series using Morlet wavelet transformation. The power spectra from the antenna with the Eccosorb AN-W-79 have the smallest magnitude, demonstrating the capacity of microwave-absorbing materials to lessen the multipath influence while not eliminating it.

## 1. Introduction

Global Navigation Satellite Systems (GNSS), particularly the Global Positioning System (GPS), have been extensively used in many earth science applications, mainly due to the significant advancement in high-accuracy positioning products [1,2,3,4,5]. The International GNSS Service [6] plays a critical role in bringing the concerted effort of the GNSS scientific community devoted to the success of this technology for its widespread use. Nevertheless, efforts are still required to further improve positioning and contribute to other GNSS-derived products. Two primary error sources remain stumbling blocks for high-precision site position solutions: tropospheric and multipath sources [7,8]. Geodesists estimate site position, troposphere and ionospheric delays assuming that there is only a direct signal as received by the GNSS receiver, but this is not attainable due to the basic design of the GNSS antenna, which receives an electromagnetic wave from multiple directions simultaneously and cannot discriminate direct from reflected directions. The multipath typically manifests as interference from the near- and far-field (Fraunhofer diffraction) reflection of the GNSS signal superimposed on the direct electromagnetic signal transmitted by the satellites. The multipath will introduce errors in the carrier phase and pseudorange measurements, and it will then propagate to other GNSS products that rely on the pseudorange and carrier phase measurements [9].

Many studies have been devoted to mitigating the effects of multipaths on site-position estimates, e.g., [8,9,10,11]. These studies found that the scattering environment within the near-field region of the antenna can introduce errors of a centimetre or more in the estimated vertical coordinate but have no discernible effect on the horizontal coordinates. For a low-cost receiver that relies on code observations, the positional error due to multipaths is very significant and can reach tens of metres [12,13], and the GNSS receiver may even experience a loss of lock in the event of severe multipath conditions [14]. Real-time kinematic positioning, even for geodetic receivers, depends on the quality of code measurements to shrink the search space in resolving the carrier phase ambiguity [15]. In doing so, multipath-corrupted pseudorange measurements can increase the time needed for initialization. King and Watson [16] demonstrated, using simulated GPS observations, that multipaths can also cause spurious periodic signals at draconitic harmonics on station coordinates. Consequently, the vertical component of the station coordinates is expected to result in erroneous tropospheric zenith delay estimates [16]. Time/frequency transfer between distant GNSS receiver clocks depends on code measurements, and the accuracy at a level of a few nanoseconds ultimately relies upon the quality of code measurements that is not adversely affected by code multipaths [17]. The multipath error also has a severe impact on the attitude determination of spacecraft where 90% of the carrier phase measurement’s overall error budget is due to multipaths [18]. Commonly, a successful reduction (“smoothing”) in pseudorange multipath depends on the exploitation of reduced multipath effects on carrier phase measurements.

A multipath is generally understood to mean that the reflected signal comes from the far-field zone, employing the geometrical ray optics principle characterising the effects [8]. The geometrical ray optics consider that the GNSS signal is only composed of direct, reflected and refracted rays. In contrast, a GNSS antenna causes currents to be produced in near-field reflecting materials, and these currents may result in considerable coupling with the GNSS antenna. This, in turn, alters the amplitude and phase properties of the GNSS antenna in comparison to its original pattern, causing the antenna to become essentially “uncalibrated” [10]. Scattering located within the near-field of the antenna is more severe than far-field scattering [8]; often, the near-field region implies the first 50 cm surrounding the antenna phase centre [19].

Typically, the influence of multipath can be analysed by linear combinations of pseudorange and carrier phase measurements for various frequencies in GNSS observables. Estey [20] used the parametrization of legacy GPS $L_{1}$ and $L_{2}$ frequencies and generated code multipaths, MP1 and MP2 metrics, respectively. However, reducing and even avoiding multipath error is much more complex. When multipaths exist, the best way to reduce the multipath effects (not in any particular order) is to install the GNSS antenna far away from potential reflecting surfaces in the antenna environment inside the Fresnel zones [21]. Normally, this is not always possible. Currently, the suppression or reduction of the multipath effect relies on an appropriately designed antenna (e.g., Choke Ring antenna ) by attenuating a GNSS signal entering the receiver [22,23]. On the receiver side, using the receiver architecture (e.g., delay-lock loop techniques [24]) can partially avoid a multipath error. While not always applied, a large ground plane located beneath the antenna also helps in modifying the shape of the antenna’s gain pattern so that it becomes insensitive to signals below and at lower elevation angles [15]. Furthermore, the GNSS antenna is designed to preferentially receive a right-hand circular polarised (RHCP) signal; therefore, the signals undergo reversal in handedness and become left-hand circularly polarized after reflecting from a nearby surface is selectively avoided. Another technique to characterize multipath effects is by repetitively observing the same satellite, looking for patterns in the carrier-phase measurements that are shifted by approximately 247 s in the case of a GPS satellite constellation, which is linked to its intrinsic one-day sidereal orbital period [25]. However, the shift between two consecutive sidereal days varies even among GPS satellites [25], and it can be even more complex for multiple GNSS constellations. In the time domain, the methods of sidereal filtering as first reported by [26] and subsequently improved by [27,28,29], or advanced sidereal filtering [30,31], have been developed to alleviate the effects of multipath errors capitalising on such repeating orbits. In addition, a slew of a new generation of GNSS signals, formed by modulation through multiplexing a wideband signal with a narrow-band signal, can improve the multipath effect [32]. Another approach is to use the signal-to-noise ratio (SNR) observations caused by the interference due to the multipath effects superimposed on the direct signal [33,34,35], isolating the multipath frequency by performing the wavelet analysis method [34,35]. Furthermore, [36] applied an advanced homomorphic (or cepstrum) filtering technique on the SNR signal to remove multipath influences and improved stochastic modelling by giving different weights to the carrier phase observations depending on their corresponding SNR values [37]. Creating multipath stacking maps is one option from long-term observation by allocating the carrier phase post-fit residuals to azimuth and elevation grid cells to mitigate unmodelled site-specific errors [38,39,40,41,42] and with further improvement in multipath stacking maps using congruent cells [43,44] and multipath hemispheric mapping [45]. Lau [46] investigated multipath mitigation using the ray-tracing technique. Park et al. [47] developed an antenna and multipath calibration system to provide *in situ* corrections, but the technique calls for a reoccupation of the site if the multipath varies. Another *in situ* technique is to place microwave-absorbing materials around the GNSS antenna; as a result, which reduces the amplitude of the reflected/scattered signals [8,10,48]. This study limits the area covered by the microwave-absorbing material around the antenna’s direct vicinity and uses GPS-only data. In spite of the multitude of studies devoted to understanding site-specific multipaths, the multipath error remains elusive, and researchers haven not yet found a “silver bullet”. Additionally, each multipath mitigation technique has its own set of weaknesses, such as implementation difficulty or incompatibility with either static or real-time navigation applications.

One strategy for minimizing multipath effects is to employ a larger-sized microwave-absorbing material that is appropriately placed in an antenna’s vicinity in an *in situ* method that avoids the need for sophisticated mitigation techniques. This paper explores how the influence of multipath on the GNSS carrier phase and code/pseudorange measurements can be reduced by using micro-absorbing materials covering a larger area than previous studies have attempted and further including other GNSS, i.e., BeiDou, GLONASS and Galileo. For this, we have established two continuously operating GNSS stations, one with microwave-absorbing materials and the other station without the Eccosorb materials. Section 2 presents the general theory on multipath detection techniques. Section 3 presents a brief theory on Fresnel zones. Section 4 outlines the experimental setup and GNSS data processing. Section 5 and Section 6 present the effect of the Eccosorb An-W-79 microwave-absorbing material on the post-fit carrier phase residual, impacts on single-point positioning, linear multipath combinations from multi-GNSS code observables, and measure the multipath frequency manifested in the SNR time series by performing a wavelet transform. A conclusion is offered in Section 7.

## 2. Multipath Theory and Signal-to-Noise Ratio

Most GNSS receivers currently use code division multiple access (CDMA) methods to split signal processing into multiple channels, allowing multiple satellites to be tracked at the same time. Most GNSS receivers use a code tracking loop or the delay-lock-loop (DLL) and the phase-lock loop (PLL) to track each satellite. These devices ensure that incoming carrier phases and codes are matched (or locked) to receiver-generated phases and codes and that the match is maintained throughout the tracking of continually received signals. The received signal is then aligned with a locally generated replica utilising a delayed lock loop. In the presence of multipaths, the signal entering the PLL becomes a composite signal between the direct signal and the reflected signal. In the GNSS receiver, the multipath manifests in many ways, such as through the additional propagation path, changes in the amplitude of the signal, changes in the phase rate akin to the additional path, the electrical-magnetic properties of the surrounding surfaces and the changes in the polarization of the signal.

To infer multipaths, geodesists use the so-called signal-to-noise ratio (SNR) records available in most GNSS receivers and reported in RINEX files. The SNR records are directly related to the carrier phase error [38]. However, different GNSS receiver manufacturers implement proprietary methodologies and definitions in recording the SNR, making it problematic to compare SNR values acquired with different receivers. In this study, we use the SNR as a proxy for the existence of a carrier phase multipath signature [49]. The dominant frequency and amplitude content of the SNR time series can be analysed to derive the qualitative measure of the multipath error that appears in the carrier phase observations. The inherent frequency content can be examined via wavelet analysis [34,35,50].

Figure 1 depicts the multipath phenomenon as received by a GNSS antenna. The radio signal arrives via two possible routes at the GNSS antenna, one through a direct route and the other (specularly) reflected from the horizontal reflector surface. Reflected signals arrive at GNSS receivers coherently or incoherently, depending on the roughness of the surface. Smooth surfaces create coherently reflected signals, which are referred to as specularly reflected signals [51]. The majority of the reflected signal occurs within the first Fresnel zone (see Section 3) about the specular point. Rough surfaces create incoherent reflected signals, which are referred to as diffusely scattered signals.

Typically, multipaths come in many forms through signal diffraction, where the incoming direct signal is diffracted at the edges of an obstruction, and most multipaths involve specular multipaths where the GNSS antenna is placed above an idealized infinite horizontal reflector surface [9]. Let us assume that there’s only one reflected signal to deal with. In this case, the direct signal $A_{d}$, the one reflected signal, $A_{r}$, and the composite signal, $A_{c}$ can be described after [49] by
(1)$$S_{c}=A_{d}\cos\phi_{d}+\alpha \;A_{d}\;\cos\left(\phi_{d}+\Delta \phi_{r}\right)$$
where $A_{d}$ and $\Delta \phi_{r}$ are the amplitude and the shifted phase of the direct signal, respectively, and α is the damping factor that relates the reflected signal $A_{r}$ with the direct signal’s amplitude, i.e., $A_{r}=\alpha A_{d}$. The amplitude $A_{c}$ of the composite signal, $S_{c}$, is estimated from:(2)$$A_{c}=A_{d}\;\sqrt{1+2\alpha \cos\left(\Delta \phi_{r}\right)+\alpha^{2}}$$

The carrier-phase multipath contribution $\delta \phi (\theta ,\alpha ,H_{o},\lambda )$ is a function of the satellite elevation angle, θ, the damping factor, the antenna height, $H_{o}$, and the wavelength (λ) of the signal
(3)$$\delta \phi \left(\theta ,\alpha ,H_{o},\lambda \right)=tan^{-1}\frac{\alpha \;sin\left(\Delta \phi_{r}\left(t\right)\right)}{1+\alpha \;cos\left(\Delta \phi_{r}\left(t\right)\right)}=tan^{-1}\frac{\alpha \;sin\left[4\pi \frac{H_{o}}{\lambda }\sin\theta \right]}{1+\alpha \;cos\left[4\pi \frac{H_{o}}{\lambda }\sin\theta \right]}$$

In general, the multipath footprint amplitudes depend on the damping factor, while the period is intimately related to the GNSS antenna height, satellite elevation and rate of change of the elevation angle of the satellite. The multipath period is longer for a short height antenna and for slowly descending or ascending satellites; a longer multipath period also occurs for close-by reflectors.

The multipath carrier phase error from Equation (Equation 3) can be converted to units of distance multiplying by λ/2π. The phase error can be expressed in units of distance for a specific GNSS,
(4)$$\delta \phi {\left(\theta ,\alpha ,H_{o},\lambda \right)}_{L}=\frac{\lambda }{2\pi }\delta \phi \left(\theta ;\alpha ,H_{o},\lambda \right)$$

The contributions of the multipath error to the GNSS coordinates following Equation (Equation 3) attain its maximum values when the multipath phase angle is $90^{\circ }$ or $270^{\circ }$. Alternatively, the maximum phase can be deduced from the amplitudes of the direct and reflected multpath signals:(5)$$\delta \phi \left(\theta ,\alpha ,H_{o},\lambda \right)=sin^{-1}\left(\frac{A_{r}}{A_{d}}\right)$$

In Equation (Equation 5), the amplitude of the direct signal can be modelled from the SNR time series for a particular satellite by fitting a polynomial. The remaining residual SNR time series largely contains the power of the reflected/scattered signal. The amplitude of the reflected signal is not easy to estimate but can be modelled using wavelet transform applied to the residual SNR (δSNR) time series ( [34,35], and see also Equation (Equation 12).

## 3. Fresnel Zones

Multipath effects are the accuracy-limiting factor in positioning applications. In this case, one can be more critical to avoid regions where significant multipaths can be expected under a particular satellite constellation. In this context, the focus is on determining active scattering regions called Fresnel zones. In radio wave propagation, the Fresnel zones concept is commonly employed. It is used to calculate diffraction and reflection losses between the transmitter and receiver. The first Fresnel zone serves as a guideline for establishing the minimum object size required to yield significant multipath signals. The term Fresnel zone is frequently used to describe the ellipsoid’s cross-section along the line of sight. The ellipsoid depicts the volume of space surrounding the line of sight that must be devoid of obstructions for clean transmission to occur. Recently, Zimmermann et al. [21] used georeferenced 3D point clouds to investigate how obstructions are identified by applying adaptive elevation masks in the antenna environment.

For a reflector with a horizontal plane and carrier wavelength λ and with satellite elevation angel, θ, and the distance of the antenna from the reflector plane, *h*, with semi-major and semi-minor axes *a* and *b*, respectively, the first Fresnel zone is given by
(6)$$b=\sqrt{\frac{\lambda h}{\sin\left(\theta \right)}+{\left(\frac{\lambda }{2\sin\left(\theta \right)}\right)}^{2}};\;\;\;\;\;\;\;\;a=\frac{b}{\sin\left(\theta \right)}$$

In general, for a flat reflector surface, the size of the Fresnel zones decreases as the satellite elevations increase independent of the satellite’s azimuth angle. The principal path from transmitting to receiving antennas is a straight line. However, it is conceivable for the signal to be reflected off objects in the antennas’ beamwidth but not in the direct route. Due to the increased length involved, whatever fraction of the transmitted signal is reflected towards the receiving antenna will arrive after the direct signal.

If the direct signal path is not obstructed and the direct and indirect signals are superimposed, checking the satellite visibility alone is not adequate. Zimmermann et al. [19,21] demonstrated a sufficient minimizing strategy to attain accuracy in the millimetre to centimetre range, especially in kinematic applications, by identifying Fresnel ellipsoids—active reflecting surfaces. In effect, the intersection of the ellipsoid with a reflecting surface produces Fresnel zones, which are active scattering regions that generate signal reflection, and, as a result, multipath effects are superimposed on the signal received on the direct signal path. For our two colocated GNSS stations that operate continuously, KBG1 and KBG2, the Fresnel zones for all azimuths, elevation angles (5, 10, 15, 30 degrees), and all satellites available within a 24 h period are colour coded and shown in Figure 2, following [52]. The experimental setups for KBG1 and KBG2 GNSS stations are discussed in greater detail in Section 4.

## 4. Experimental Setup and Data Acquisition

Two continuously colocated GNSS stations (KBG1 and KBG2) were constructed in August 2020 in the city of Luxembourg, Kirchberg campus, on the roof of the JFK building (6.2${}^{\circ }$ N, 49.6${}^{\circ }$ E, 404.9 m). The two GNSSs are equipped with Trimble Alloy GNSS reference receivers. These receivers are attached to a Trimble geodetic Choke Ring GNSS antenna with IGS naming, TRM159800.00, integrated with a radome-type SCIS. The antennas are mounted on a 3.5-m aluminium mast (LECLERC type). The two sites use a common receiver/antenna and only differ in their serial number at the site, with KBG1 having SCIS-5850337009 and KBG2 having SCIS-5628351662. The two Trimble antennas have also been individually calibrated in a clean environment within an anechoic chamber of the University of Bonn for phase centre offset (PCO) and phase centre variation (PCV) models [53]. The model values are provided in ANTEX version 1.4 format. The anechoic chamber antenna calibration provides models for 25 frequencies for BeiDou (C01, C02, C05, C06, C07, and C08), Galileo (E01, E05, E06, E07, and E08), GLONASS (R01, R02, R03, R04, and R06), GPS (G01, G02, and G05), QZSS (J01, J02, J05,and J06) and SBAS (S01 and S05). The two stations are located in challenging locations but very close to each other with a distance of 5 m, as shown in Figure 3, which elucidates the experimental setup; to the left is station KBG1, and to the right is station KBG2. The station KBG1 antenna is mounted on a single three-metre LECLERC aluminium mast with four-legged three-metre LECLERC aluminium masts supporting the wooden frame carrying the Eccosorb material.

As indicated in the introduction, one strategy to potentially mitigate the consequences of signal scattering is to cover the scatterer with microwave-absorbing material. The KBG1 station is equipped with a lightweight, flexible, polyurethane foam broadband microwave-absorbing material, specifically an Eccosorb AN-W-79 sheet sealed with neoprene-coated nylon fabric. A single Eccosorb has 11.4 cm nominal thickness polyurethane sheets cut into 61 cm × 61 cm sections. It consists of nine sections with a total area of 3.35 m${}^{2}$ surrounding the perimeter of the KBG1 antenna, with the individual unit weighing 2.95 kg. For fairly wet and high humidity environments, a sealed version of Eccosorb AN-W-79 is preferable, providing improved outdoor properties. This particular Eccosorb material reflects less than −17 dB of normal incident energy above 600 MHz, which is below the signal frequencies that operate the GNSS signals. Figure 4’s left and middle depict an image of an Eccosorb AN-W-79 covered by an olive green colour attached to the perimeter of the KBG1 station antenna, whereas the antenna for station KBG2 without the Eccosorb material is shown on the right of Figure 4. The Eccosorb material should block scattering or diffraction at least within 90 cm from the antenna from any direction, including the wooden frame and the Choke Ring of the antenna. This means that the scatterer located in the near-field of the antenna has negligible electromagnetic coupling with the antenna. Another GNSS antenna next to the KBG1 GNSS station is also installed with a Trimble TRM159800.00 Choke Ring GNSS antenna, which is a radome (type SCIS) mounted on a single three-meter LECLERC Aluminium mast, as shown in Figure 3, to the right.

The two stations have collected multi-GNSS data continuously since 20 September 2020 (day of year (DoY) 264). The data help us assess the characteristics of multipath effects with two configurations, i.e., with and without the Eccosorb material. The two stations are located where many obstructions are prone to multipath effects. The GNSS data collected between 1 January 2021 (DoY 1) and 5 May 2021 (DoY 125) were used for this study. The multi-GNSS observables were collected with zero elevation angles. The multi-GNSS data were retrieved every 1 s and 30 s epoch in daily RINEX 3.04 files containing raw, dual-frequency pseudorange and carrier phase observations. The post-fit residuals of the two stations were estimated using the GAMIT/GLOBK software package (Ver 10.7) as part of a regional network in a double differencing strategy; see the details in Section 6. We applied all four GNSS observations with an elevation cut-off angle of $3^{\circ }$ together with high-precision GNSS orbits and clock products from the centre for orbit determination in Europe (CODE) in the IGS14 reference frame [54]. Additionally, essential input data from CODE, such as Earth orientation parameters, eclipse shadow events, and external products such as ocean tidal models, were employed. We used the tidal constituents’ coefficients of the FES2004 model [55]. For neutral atmospheric delay modelling, we used VMF1 gridded map products from [56] that include both dry and wet delays and coefficients of the mapping functions that provide the variability of the atmospheric delay over the whole elevation range. The VMF1 grid maps were constructed from the European Centre for Medium-Range Weather Forecasts (ECMWF) weather model based on global meteorological data. We also corrected the carrier phase observations for the ionosphere effects through the so-called mathematical linear combinations, eliminating the first-order ionosphere contributions. The higher-order calibrations using CODE global ionospheric maps for the remaining ionosphere effects in combination with the new generation magnetic field model, IGRF12, were implemented. The individual antenna calibration models were used. We also estimated the ambiguity of integer values depending on the baseline length. Furthermore, the atmospheric parameters were estimated every 2 h, including the zenith wet delay and gradient parameters in the east-west and the north-south components, to model azimuthal asymmetries. The zenith wet was constrained delay at 5 cm per square–root hour between the epochs and for the two gradients at 5 mm per square-root hour. We also processed the two close-by stations using GipsyX v1.7 software [57] in a precise point positioning (PPP) strategy. We included all the latest satellite orbit and high rate clock products from JPL. The post-fit residuals from the two strategies produced a consistent estimate.

## 5. Impacts of the Microwave Absorption: Observation Results

### Multi-GNSS Code Multipath

The multipath effect is present in both the code and carrier phases of GNSS measurements; however, the code multipath is substantially larger and more variable between the receiver and antenna types. For the single point positioning technique, the multipath error is a significant contributor to the accuracy of the code measurements. For proper observation weighting, the understanding of the multipath effect and code noise can be useful [58]. Additionally, this data may reveal unique features of the receiver or information about the station’s surroundings. Standard dual-frequency precise point positioning (PPP) solution performance depends on the pseudorange observable and code multipath error quality. Consequently this affects the time it takes to fix the float ambiguity to an integer. Furthermore, the presence of cycle slips contributes to resolving integer ambiguities and multipath estimation.

In this study, we computed the code multipath on the pseudorange, $MP_{k}$ for all frequencies and for all accessible signals, including multi-GNSS that provides dual-frequency observations in a way that allows the code multipath to be isolated. Since this technique is computationally simple, it has been widely used to measure total code multipath and noise on frequency *k* as follows:(7)$$MP_{k}=P_{k}-\frac{\left(f_{j}^{2}+f_{k}^{2}\right)}{\left(f_{i}^{2}-f_{j}^{2}\right)}\frac{f_{i}^{2}}{f_{k}^{2}}L_{i}+\frac{\left(f_{i}^{2}+f_{k}^{2}\right)}{\left(f_{i}^{2}-f_{j}^{2}\right)}\frac{f_{j}^{2}}{f_{k}^{2}}L_{j}+C_{avg}$$
where *i*, *j* and *k* are frequency indices, $L_{i}$ and $L_{j}$ are carrier phase observables for frequencies $f_{i}$ and $f_{j}$ for the same satellite (i≠j) and $C_{avg}$ is the bias, i.e., constant hardware delays and constant ambiguity. For a given satellite and receiver pair, the bias corresponds to fixed numbers across the whole track. The $C_{avg}$ is computed using a moving average technique, i.e., a moving average window of $MP_{k}$ estimates over a sequence of consecutive epochs of a specified duration. When estimating Equation (Equation 7), we implicitly disregard multipath from carrier phase observables, that are often two orders of magnitude smaller than the code multipath [23].

For modernized and new GNSS constellations with multi-frequency signals, a number of code multipath combinations are possible. We form GPSM1C, GPSM1X, GPSM2W, GPSM2X and GPSM5X for GPS; GLOM1C, GLOM1P, GLOM2C and GLOM3X for GLONASS; GALM1X, GALM5X, GALM6X, GALM7X, and GALM8X for Galileo; and BDSM1X, BDSM2I, BDSM5X, BDSM6I, and BDSM7I for BeiDou. For the detailed signal observation type, frequency and tracking channel numbering, see (https://files.igs.org/pub/data/format/rinex304.pdf, accessed on 20 December 2021).

We estimated the multipath root-mean-square (RMS) using the Anubis [58] software package over a sequence of 30 s epochs using all the linear combinations of available frequency bands for each GNSS constellation using Equation (Equation 7). The computation of the pseudorange multipath combinations within each GNSS constellation assumes that the biases are held constant only when there are no cycle slips while tracking the GNSS signals, so in practice, the mean component of the multipath combination is removed; as a result, only the RMS variations are reported [20].

Figure 5 shows the code multipath for the two GNSS stations KBG1 (top) with Eccosorb and KBG2 without Eccosorb (bottom) for all possible linear combinations. The RINEX 3.04 format is used to produce this analysis employing all the available frequency bands for January–May 2021. In general, the code multipath shows a steady estimate with some small minor variabilities in GPSM1X and on all BeiDou multipath linear combinations. In general, BeiDou signals performed the worst for both stations and GLONASS M1C, and M2C showed the next worst performance, followed by GPS M1C. Galileo signals showed the best performance with a particular best performance from Galileo M8X for both stations due to a wide-band alternate binary-offset-carrier (AltBOC) modulation [32]. A slight variation exists with the other multipath linear combinations within the Galileo frequency bands, with slightly M1X multipath linear combination lower performance, again depending on the modulation used. The station KBG1 with Eccosorb material shows a much better performance than the KBG2 with no Eccosorb material. The performance of Galileo stations is much better for station KBG1 with Eccosorb compared to KBG2 station. On average for Galileo, the signal performs more than 60% better for the station with Eccosorb, KBG1. For GLONASS, particularly for M3X and M1P multipath combinations, the performance improved up to 50%. For BeiDou, the improvement is more than 30%; for GPS the improvement reaches 40 %; see Figure 6 for the relative improvement.

For high-precision positioning applications, detecting cycle slip is crucial. Carrier-phase measurements are susceptible to cycle slips, causing a bias in GNSS observables and, if undetected (and uncorrected), affecting the estimated station coordinates. For real-time kinematic solutions, cycle slips must first be identified and repaired to aid faster ambiguity resolution and improve positioning accuracy. Typically, a multipath causes the degradation of receiver tracking capabilities and potentially introduces cycle slips in the recorded GNSS observables. Figure 7 depicts the number of identified carrier-phase cycle-slips during a continuous phase tracking for colocated stations for the period January 2021 until the 5 May 2021 using the Trimble Alloy reference receiver. Figure 7a shows the cycle slips for the four GNSS constellations, i.e., BeiDou, Galileo, GLONASS and GPS, for carrier-phase observables. Clearly, the carrier-phase tracking shows very few cycle slips for the two close-by stations under investigation, with BeiDou and GLONASS performing worse. Galileo and GPS perform much better. The carrier phase cycle slips for BeiDou L1X and L5X and GLONASS L3X show very few cycle slips, as does the Galileo carrier phase signals; L1X (E1), L5X (E5a), L6X (E6), L7X (E5b) and L8X (E5a + b) and the GPS carrier phase signals; and L1X, and L2X as recorded by Trimble Alloy reference receivers. While the cycle slips for GPS carrier phase signals for L2X, L1C and L2W are relatively small, less than 50, the small number of cycle slips using Eccosorb material is appreciable. Figure 8 shows an improvement in percentage using Eccosorb material compared to the station without the Eccosorb. The Galileo carrier phases show no change. A gain for BeiDou carrier phase signals (L2I, L6I, L7I), for GLONASS (L1C, L1P, L2C), and for GPS (L1C, L2W, L2X) show an improvement between 15% and 60% when using Eccosorb microwave-absorbing material.

## 6. Effects of Multipath on GNSS Carrier Post-Fit Residuals

In contrast to the pseudorange measurements, the multipath error on the carrier phase observations is limited to a quarter of the wavelength of the GNSS signal. Granstrom [40] used the carrier phase post-fit residuals as a useful tool for examining the influence of multipath effects. The elevation dependence of the phase residuals, averaged across all azimuths, provides a useful metric of the phase modelling and multipath effects at each site. We compared the residuals of linear combinations of dual frequencies of GLONASS, Galileo and GPS carrier-phase measurements employing the ionospheric-free (LC) linear combination using GAMIT [59] and GIPSY [57] software packages.

This results in post-fit residuals that contain multipath errors from individual signals and other unmodelled errors. We want to emphasise that we have applied an individually calibrated PCO and PCV antenna model, making its contributions to the total post-fit budget insignificant. The post-fit carrier phase residuals for station KBG1 as a function of elevation angle (between $3^{\circ }$ and $90^{\circ }$) with Eccosorb material are shown in Figure 9a,c,d and similarly for site KBG2, without Eccosorb material, is shown in Figure 9b,e,f for all satellites from 1 March 2021. The residuals show large oscillations at low elevation angles to the satellite with respect to the local horizon, with a particular multipath error signature with higher magnitude and unmodelled atmospheric delay. For all the multi-GNSS ionospheric-free linear combinations, we formed for GPS using L1/L2, GLONASS G1/G2, and Galileo E1/E5a frequencies.

Site KBG1 shows a lower RMS than the KBG2 station for all three multi-GNSSs considered. Figure 9a shows the residuals from the GAMIT solutions for Galileo, Figure 9c for GLONASS, and Figure 9d for GPS for the station KBG1. The RMS shows a clear demonstration that the Eccosorb material reduces the multipath error from 14.15 mm to 12.27 mm for Galileo, from 16.58 mm to 15.25 mm for GLONASS, and from 16.33 mm to 14.83 mm for GPS. Hence, there is a relative reduction in RMS 13% for Galileo and 9% for GLONASS and 10% for GPS. The Eccosorb material clearly shows a reduction in multipath signature but did not wholly eliminate the effect. Ning [48] also shows an improvement using Eccosorb material but only slightly, likely due to the size of the Eccosorb that surrounds their GNSS antenna, which is nine times less than this study has used. We have also used the PPP solutions applying nearly identical products using the GipsyX software package to evaluate the post-fit residuals (not shown). The two software post-fit residuals are consistent with each other, again showing smaller multipath footprint for station KBG1 with the Eccosorb materials than KBG2 without.

Furthermore, we computed the RMS of the post-fit carrier phase residuals for a $1^{\circ }$ bin width for an elevation angle between 3${}^{\circ }$ and 90${}^{\circ }$ for the KBG1 and KNG2 stations, as depicted in Figure 10. The RMS value decreases by approximately about 1 mm for elevation angles between $3^{\circ }$ and $35^{\circ }$; after that, the effect of the Eccosorb material becomes insignificant, and the two stations show similar RMS values.

### 6.1. Single Point Positioning

The two stations (KBG1 and KBG2) are continuously operating with all four constellations recorded at 1 s and 30 s epochs. Typically, positioning from pseudorange measurements with broadcast ephemeris is possible when at least four satellites are visible in the sky. At our latitude, at least seven satellites are observed from any of the constellations. Single-point positioning (SPP) can readily be available using only psuedorange measurements at every epoch or once every 24 h. In GNSS processing, the carrier phase ambiguity resolution is generally initialized using pseudorange measurements; multipath-corrupted observations can consequently take longer to resolve the ambiguities to integers [13]. Figure 11 and Figure 12 present the single-point positioning evaluated for multi-GNSS constellations with and without Eccosorb materials as explained in Section 2. The site-specific multipath shows an effect on the position qualities from pseudorange observations looking at the uncertainty derived in north-south (NS) and east-west (EW) directions for all four GNSS constellations employed here. For station KBG1 with Eccosorb, the standard error for GPS in the NS direction is 1.7 m, while in the EW direction, it is 1.3 m; for Galileo, it is 0.7 m and 0.5 m respectively, highlighting the much better accuracy achieved by Galileo. For GLONASS, the NS and EW standard errors are 5.9 m and 4.0 m, respectively, with the lowest accuracy coming from BeiDou at 7.4 m and 7.7 m, respectively. We also evaluated the instantaneous accuracy of SPP for the close-by station, KBG2 without Eccosorb. The standard error in the NS direction for GPS is 2.0 m and 1.7 m in the EW direction; for Galileo, it is 0.9 m and 0.9 m; for GLONASS, the NS and EW standard errors are 7.6 m and 4.0 m; and the lowest accuracy comes from BeiDou, with 7.1 m and 8.0 m, respectively. The reductions in standard errors are significant for all the multi-GNSS-based SPP solutions for the station with microwave-absorbing material. For GPS, the improvement is between 15 and 23%; for Galileo, it is between 22% and 45%; GLONASS is 22%; and BeiDou is about 4%. In addition, for the station KBG1, we computed a combined GPS + GLONASS + Galileo + BeiDou solution with an NS direction accuracy of 1.7 m and an EW direction accuracy of 1.4 m. Similarly, for the KBG2 station, the combined GPS + GLONASS + Galileo + BeiDou solution shows an accuracy of 2.0 m and 1.6 m in the NS and EW directions, respectively. The Eccosorb material improves the single-point positioning solution accuracy by 12 to 15% for the combined solution.

The microwave-absorbing material Eccosorb AN-W-79 shows convincing results in reducing the noise level of the code measurements.

### 6.2. Measurement of SNR Frequency Content Using the Wavelet Transform

GNSS signals have the properties of nonstationarity characteristics, which vary over time. As constituents of the GNSS signal recorded in the GNSS receiver, multipaths also shows a nonstationarity process. To understand and localize the footprint of multipaths, the customarily applied Fourier analysis is not ideal in detecting and localizing the dominant frequency in a signal. This is because the Fourier transforms typically capture the global features of the signal due to the Fourier basis functions, i.e., sines and cosines. Hence, no local features of the signal are captured. We have applied wavelet analysis strategies to measure the multipath frequency and power content of the residual $SNR_{n}$ time series with a sampling interval at δt and n=0,...,N−1. Wavelet analysis may be more suited than classic Fourier analysis by revealing a time series in terms of frequency and time information, which is particularly advantageous when the signal is nonstationary. GNSS multipath is one of the signal types that has nonstationary properties [34,35,60], since its frequency changes with the rate of change of the satellite elevation angle. The purpose of wavelet spectra estimation is to elucidate the periodic multipath error and, in so doing, estimate how this multipath error affects GNSS positioning. We use the continuous wavelet function $\psi {\left(S\right)}_{o}$ which satisfies with zero mean the localization at both times and frequencies [61], and integrating the wavelet basis over a certain time becomes close to zero. One of the wavelet functions that satisfies the condition of locality is a Morlet wavelet shown in Equation (Equation 8), a wavelet produced by modulating a complex sine wave by a Gaussian function. In this analysis, we heavily adapted the wavelet analysis strategies used in [50] and recently expanded by [34,35].
(8)$$\psi \left(x\right)=\pi^{-1/4}e^{i\omega_{o}x}e^{(-x^{2})/2}$$

The continuous wavelet transform $W_{n}$, also a complex function, of a discrete sequence the residual SNR, i.e., $\delta SNR_{n^{\prime }}$, is defined as the convolution of $\delta SNR_{n^{\prime }}$ with a varying scale and translating along the time index of the Morlet wavelet basis ψ(x):(9)$$W_{n}\left(S\right)=\sum_{n^{\prime }=0}^{N-1}\delta SNR_{n^{\prime }}\psi *\left[\frac{(n^{\prime }-n)\delta t}{s}\right]$$

Once the wavelet function is selected, we can compute the wavelet transform for a set of scales to form a localized frequency and amplitude of the δSNR time series, where (∗) is a complex conjugate operator. It is convenient to compute the wavelet transform at different scales expressed as a fractional power of two,
(10)$$S_{j}=s_{o}2^{j\delta_{j}},\;\;\;\;\;j=0,1,...,J,\;\;\;\;\;\;\;\;\;\;\;\;\;\;\;\;\;\;\;\;\;\;J=\delta j^{-1}log_{2}\left(N\delta t/s_{o}\right)$$
where $s_{o}$ is the smallest resolvable scale, approximately twice the sampling interval, and *J* is the largest scale. For the Morlet wavelet, the choice, δj, depends on the width of the spectral space of the wavelet, and we adopted the same value as [50] at 0.125.The Morlet continuous wavelet transform, which is implemented in the Python package, was utilized *pyCWT* [50,62] with the power spectral density correction of [60].

The result of the convolution between the Morlet wavelet and the residual SNR will help in the construction of the δSNR showing the amplitude of any features with respect to scale and how this amplitude also changes with time; i.e., $|W_{n}\left(S\right)|$ is the amplitude, and the phase can be constructed from both the real and imaginary part of the $W_{n}\left(S\right)$ and the wavelet power spectrum provided as $|W_{n}{\left(S\right)|}^{2}$. Transforming the residual SNR time series using wavelet transform reveals the temporal features or dynamics and extracts spectral specific (frequency) content of the signal over time.

The raw SNR was filtered for zero-value occurrences, with sample 1 Hz data including significant multipath peaks, as demonstrated in this example in Figure 13. Before we applied the wavelet representations to the SNR times series, we separated the SNR contributions to the direct signal and the residual SNR components to enhance the spectral analyses of the SNR more robustly by demeaning the time series. The steadily changing trend in the SNR represents the direct signal. Typically, the residual SNR is estimated by fitting a polynomial to the recorded SNR values at the two close-by stations on 1 March 2021 for GPS satellite PRN27, selected according to its Fresnel footprint; see Figure 2. Our choice of the polynomial fit is based on the minimum RMS between the model polynomial to the raw SNR. Figure 13 depicts how to isolate the quasi-oscillation—a proxy to the existence of multipath by removing a polynomial fit to the raw SNR time series. This quasi-oscillation is more prominent for station KBG2, which has no Eccosorb material, compared to station KBG1, which has Eccosorb material; see Figure 13.

We show the wavelet power spectra of the δSNR to study the multipath contributions closely due to the specific obstructions to the two GNSS stations shown in Figure 3 and Figure 4. The SNR data are typically reported on a logarithmic scale (dB-Hz); here we converted to a linear scale using $10^{\left(SNR[db-Hz]/20\right)}$.

Following Equations (Equation 9) and (Equation 10), we examined the properties of δSNR over a range of scales (frequencies). One can define the scale-averaged wavelet power between two wavelet scale bands, $s_{1}$ and $s_{2}$, as:(11)$$W_{n}^{2}=\frac{\delta j\delta t}{C_{\delta }}\sum_{j=1}^{j=2}\frac{|W_{n}\left(s_{j}\right){|}^{2}}{s_{j}}$$

The constant factor $C_{\delta }=0.776$ is chosen for the Morlet mother wavelet, as reported in [50]. The amplitude of the δSNR, i.e., $A_{r}$ (compared with Equation (Equation 5)) for selected scale bands within a certain time period range can be estimated taking the square root of Equation (Equation 11) over all scales:(12)$$A_{r}\left(t\right)=\sqrt{W_{n}^{2}}=\sqrt{\frac{\delta j\delta t}{C_{\delta }}\sum_{j=0}^{j=J}\frac{|{\left(W_{n}s_{j}|\right)}^{2}}{s_{j}}}$$

Figure 14 shows the average power of the residual SNR over all scales, applying Equation (Equation 12) between different period bands, i.e., 2–20, 20–40, 40–80, and 80–120 s, for two consecutive DoYs 060 and 061 (1 and 2 March 2021). Figure 14, which shows the scale-average of the wavelet power as a function of time, shows distinct significant wavelet power. In all cases, station KBG2 manifests higher power than KBG1. In the 80–120 period band, the power of KBG2 (with no microwave-absorbing material attached to it) shows a higher power than KBG1 for both DoYs. A similar wavelet power is also manifested for period bands 40–80 again for station KBG2 for the time series between 61,000 to 61,200 s and then back again to a diminished power for the time series between 62,000 to 62,850 s (for 850 s). Slightly higher power is shown for the period band 20–40, which is fairly short-lived and stays for 200 s. For the period band between 2 and 20, KBG2, in general, show a higher power but is very short-lived, mostly appearing at the end of the time series. Similarly, KBG1 also showed some power at the end of the time series for the period band 2–20. What is also evident for station KBG2 is that near-identical higher wavelet power is manifested from the two consecutive DoYs. The wavelet power appeared approximately 240 s earlier for DoY 61, confirming the sidereal repeatability nature of the multipath, and which is associated with the location of the KBG2 antenna in relation to the reflecting surfaces.

Figure 15 shows the spectrum of the Morlet wavelet transform on the δSNR time series on 1 March 2021 (DoY 60) on the left from station KBG1 and to the right from the same date for station KBG2. For this study, we chose GPS satellite PRN27 as the satellite that ascends a subtending angle between 255${}^{\circ }$ and 275${}^{\circ }$ in azimuth from elevation angles $2^{\circ }$–$30^{\circ }$. For station KBG1, a significant power was observed for a period of 60–100 s but manifested for 260 s. During this, the satellite moved one degree in azimuth (from 265${}^{\circ }$ to 266${}^{\circ }$) and ascended from ∼16${}^{\circ }$ to 18${}^{\circ }$ in its elevation angle. For station KBG2, the same satellite shows a significant spectral power for the period 30–64 s spanning ∼100 s. During this, the satellite moved one degree in azimuth (from 258${}^{\circ }$ to 259${}^{\circ }$) and ascended from 7 to 7.5${}^{\circ }$. Again, KBG2 shows a significant power for the period 64–128 s but spans much longer (for ∼850 s), though at half the power of earlier. Its power was concentrated between 12.5${}^{\circ }$ and 17.7${}^{\circ }$. At this stage, the satellite moved further to the south-southwest. The significant δSNR power occurring at periods 0–128 s does not correspond to the ideal horizontal reflecting the infinite extent surface that corresponds to the height of the antenna at 3.5 m above the ground. Therefore, the various structures on the roof of the building further from the GNSS antenna contribute to the multipath budget, more adversely affecting station KBG2 without the Eccosorb material.

## 7. Conclusions

Multipaths have remained one of the dominant sources of errors that limit site position accuracy and the atmospheric propagation delays estimated from GNSS receivers that employ carrier phase and code observables. No standard model or technique has been developed yet. In this study, we devised a *in situ* strategy that avoids any complex modelling of the multipath effect but rather limits the impact of multipath in the first place, contaminating the carrier phase and code observables. We have established two close-by, continuously observing multi-GNSS stations under challenging conditions, with different metal frames, poles, various sizes and shapes, ensuing a complex multipath environment. The two configurations were used: one with and one without Eccosorb AN-W-79—a specialized electromagnetic signal absorption material—placed around the antenna. The station with the *in situ* microwave-absorbing material has shown a significant improvement in the accuracy of the single-point positioning that depends on the code measurements. The reductions in the standard error are between 4% and 45% for all four GNSSs (BeiDou, GLONASS, Galileo and GPS). In addition, we computed the combined GPS + GLONASS + Galileo + BeiDou single-point positioning solution with a reduction in the standard error of up to 15%. Furthermore, the code multipath linear combinations using the various frequencies from the GNSS, on average, show an improvement from 20% to 70%. All the GPS-based code multipaths, i.e., GPSM1C, GPSM1X, GPSM2W, GPSM2X and GPSM5X, show an improvement of 20–40%. In particular, the improvement of GPSM5X with the C5 code using the I + Q channel reaches more than 40%. For GLONASS, the same pictures appear. The linear code combinations, such as GLOM1C, GLOM2C, GLOM2C and GLOM3X, show improvements of between 20% and 55%; for Galileo, the code multipath combinations GALM1X, GALM5X, GALM6X, GALM7X, and GALM8X show the largest improvement, reaching 60–70%; and for BeiDou, combinations BDSM1X, BDSM2I, BDSM5X, BDSM6I, and BDSM7I reaches 30–40%. The cycle slip detection and subsequent repair, particularly in a dynamic scenario, are complex, impending the full exploitation of the otherwise precise carrier phase measurements, and this study shows a nearly 20–60% improvement for BeiDou GLONASS and GPS. The Galileo signal shows no improvement.

The use of *in situ* microwave-absorbing material shows convincing results in reducing the code multipath noise level and in carrier phase cycle slip. The quality of code measurements will thus lead to shrinking the search space in resolving the carrier phase ambiguity [15]. Furthermore, multipath has shown to severely impact the attitude determination of spacecraft [18]. While the *in situ* method was evaluated with a static antenna, microwave-absorbing material can be used for kinematic applications as well.

The residual signal-to-noise ratio (SNR) shows pronounced oscillations for the station (without Eccosorb). The time-varying multipath amplitude of the residual SNR for the station equipped with microwave-absorbing material shows consistently much more reduced magnitude, confirming a limited multipath footprint in the carrier-phase observations. While individual satellite SNR values show a diminished power in the station with Eccosorb, the error involving carrier phase residuals also show a reduced noise level—a confirmation from processed GNSS observables in the form of post-fit residuals. The multipath frequency content obtained by performing Morlet wavelet analysis on the residual SNR shows the complexity of the frequency content in the environment from our continuous GNSS stations. While the multipath footprint is prevalent, particularly at lower elevation angles, the use of Eccosorb material reduces the effect but does not wholly remove it. Therefore, the next phase of our research will employ the multipath stacking maps using the raw post-fit carrier phase residuals to further mitigate the multipath error that is more evident at lower elevation angles.

## Figures and Tables

**Figure 1 sensors-22-03384-f001:**
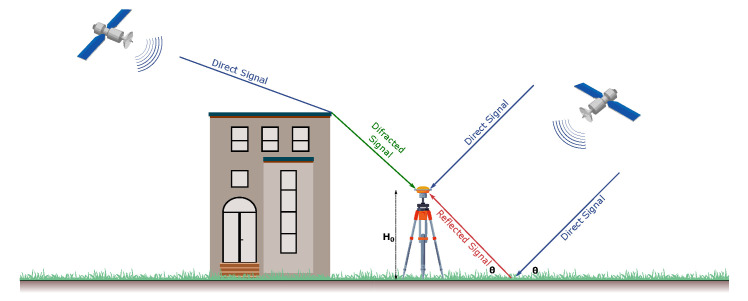
Multipath geometry. The multipath is the interference between the direct and the reflected signal shown in the idealized horizontal surface. H is the height of the GNSS antenna.

**Figure 2 sensors-22-03384-f002:**
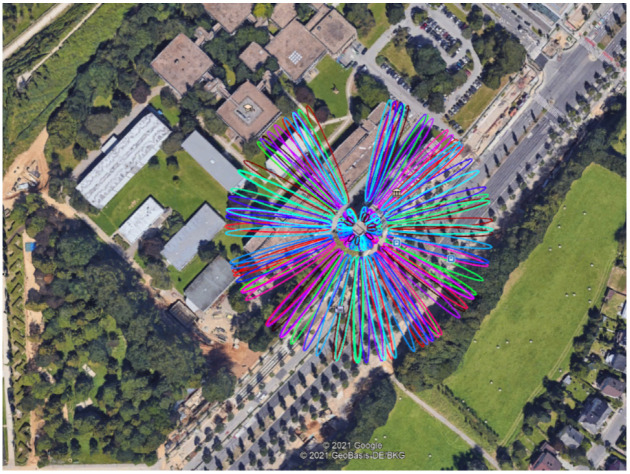
The Screenshot of the first Fresnel Zone at KBG1/KBG2 on the roof of the JFK building, Kirchberg campus, Luxembourg city, using elevation angles of 5, 10, 15, and 30 degrees. The coloured ellipses represent all GPS satellites visible at KBG1/KBG2 location. An antenna height of 3.5 metre is used. Credit of background image: Google Earth.

**Figure 3 sensors-22-03384-f003:**
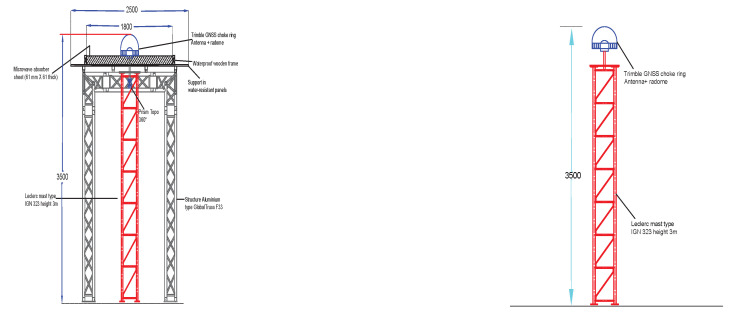
Autocad drawing of GNSS station KBG1 to the (**left**) and KBG2 to the (**right**). All the values are given in millimetres.

**Figure 4 sensors-22-03384-f004:**
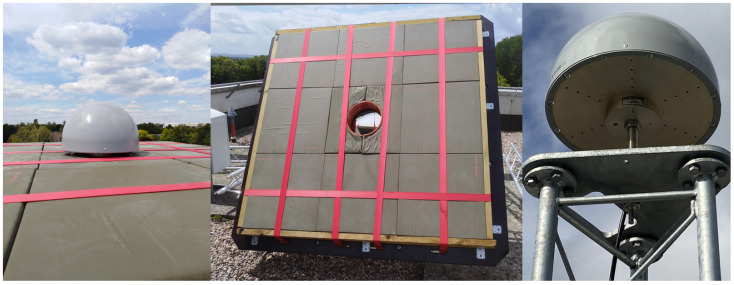
The photograph of KBG1 station’s Trimble (TRM159800.00) geodetic Choke Ring GNSS antenna integrated with SCIS radome (**left**) and the Eccosorb AN-W-79 (microwave absorber) material that surrounds station KBG1 (**middle**) and KBG2 antenna, Trimble (TRM159800.00) geodetic Choke Ring GNSS antenna integrated with an SCIS radome (**right**).

**Figure 5 sensors-22-03384-f005:**
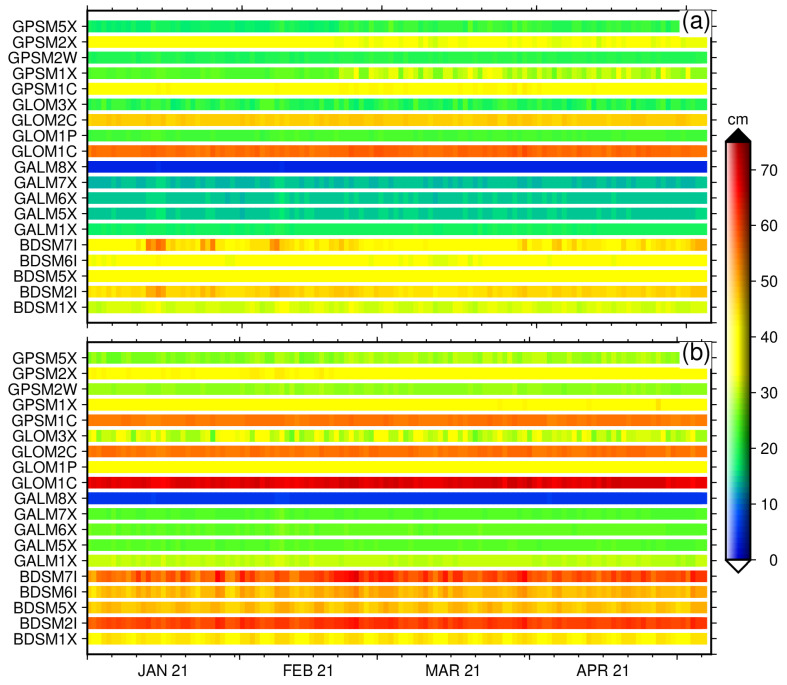
Pseudorange (Code) multipath estimated for all possible linear combinations as recorded at stations KBG1 (**a**) and KBG2 (**b**) during January–May 2021.

**Figure 6 sensors-22-03384-f006:**
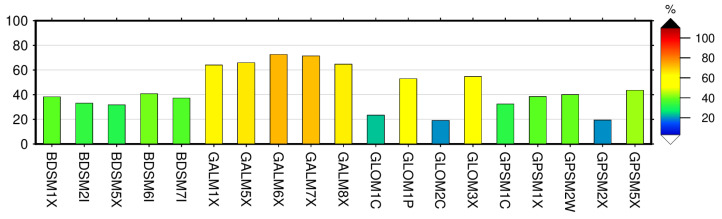
Improvement of pseudorange (Code) multipath estimated for all possible linear combinations as observed at station KBG1 with Eccosorb with respect to KBG2 without Eccosorb averaged over the period January–May 2021.

**Figure 7 sensors-22-03384-f007:**
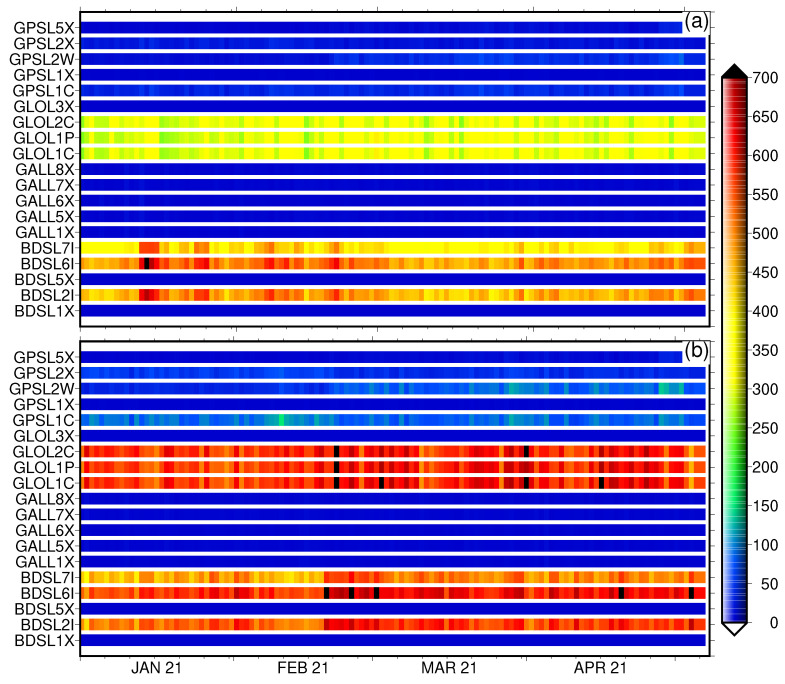
The number of identified real phase cycle-slips during continuous phase tracking for station KBG1 with Eccosorb (**a**) and for station KBG2 without Eccosorb (**b**) for the period January 2021 until the 5 May 2021 using state-of-the-art Trimble Alloy reference receivers.

**Figure 8 sensors-22-03384-f008:**
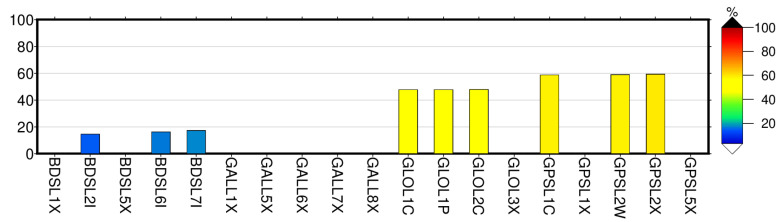
The number of cycle slips improved using the Eccosorb material with respect to not using the Eccosorb microwave-absorbing material. The improvement percentage is an average value for the period January 2021 until the 5 May 2021 using state-of-the-art Trimble Alloy reference receivers.

**Figure 9 sensors-22-03384-f009:**
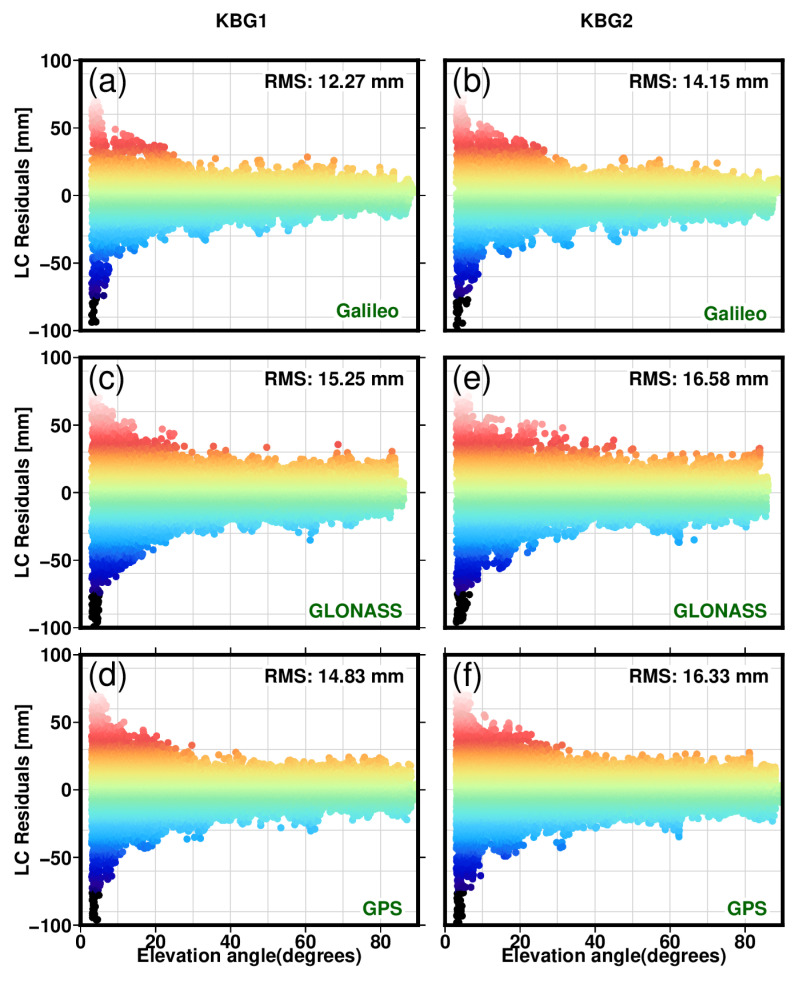
Post-fit LC carrier-phase residuals for individual GNSS constellations, i.e., Galileo (**a**), GLONASS (**c**) and GPS (**d**) for station KBG1 with Eccosorb and Galileo (**b**), GLONASS (**e**) and GPS (**f**) for station KBG2 without Eccosorb. The dataset was acquired on 1 March 2021.

**Figure 10 sensors-22-03384-f010:**
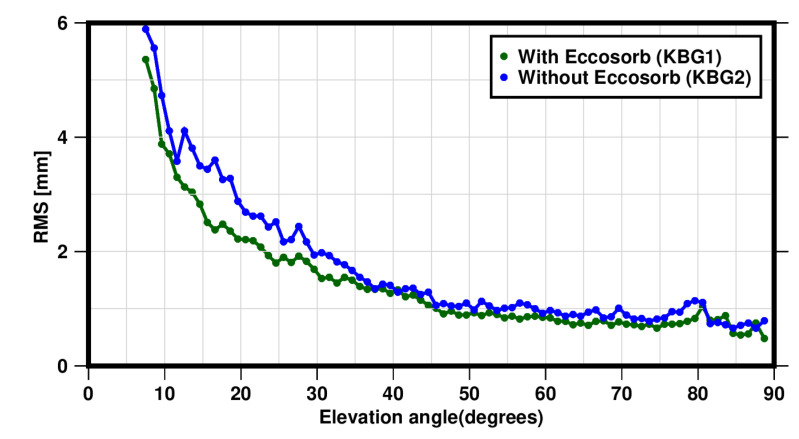
The RMS values of the post-fit carrier phase residuals (LC) of the L1 and L2 phase observables for station BKG1 with the microwave absorber in blue solid colour and for station BKG2 in dark green solid colour for elevation angles with a bin width of $1^{\circ }$.

**Figure 11 sensors-22-03384-f011:**
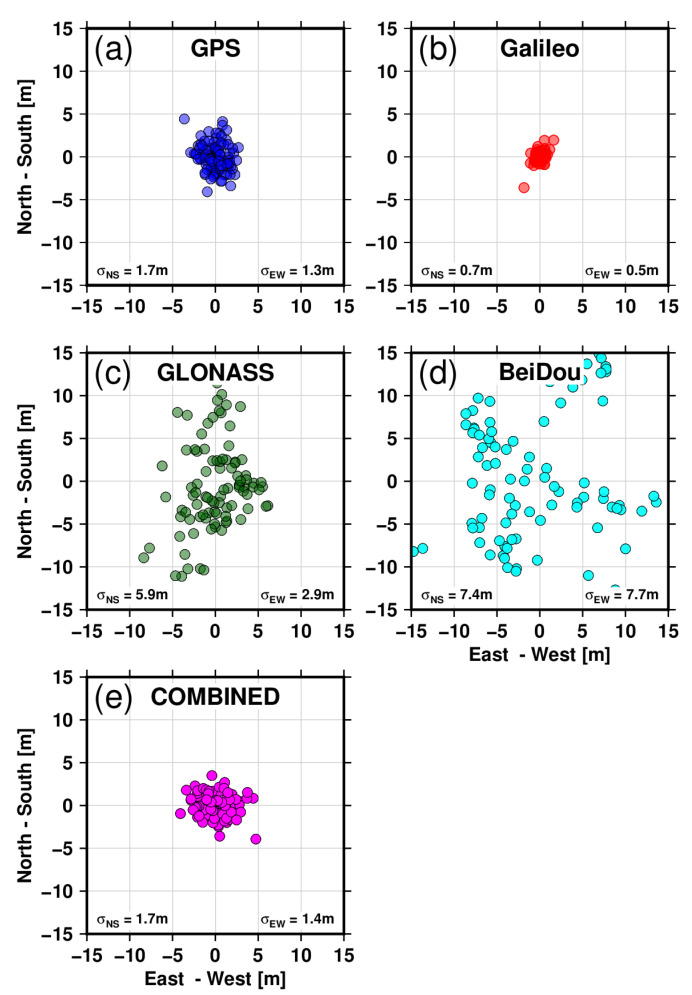
Single-point positioning for site KBG1 with Eccosorb absorbing material surrounding its antenna using multi-GNSS observables. Results are shown in blue for GPS (**a**), in red for Galileo (**b**), in dark green for GLONASS (**c**) and cyan for BeiDou (**d**), and in the magenta for COMBINED (**e**). The positioning accuracy using GPS in the north-south direction is 1.7 m, and in the east-west direction, it is 1.3 m; for Galileo, it is 0.7 m and 0.5 m, respectively, highlighting the added value of Galileo; for GLONASS, it is 5.9 m and 4.0 m; and the lowest accuracy comes from BeiDou, with 7.4 m and 7.7 m. We have also derived a combined GPS + GLONASS + Galileo + BeiDou solution with a north-south direction accuracy of 1.7 m and east-west accuracy of 1.4 m.

**Figure 12 sensors-22-03384-f012:**
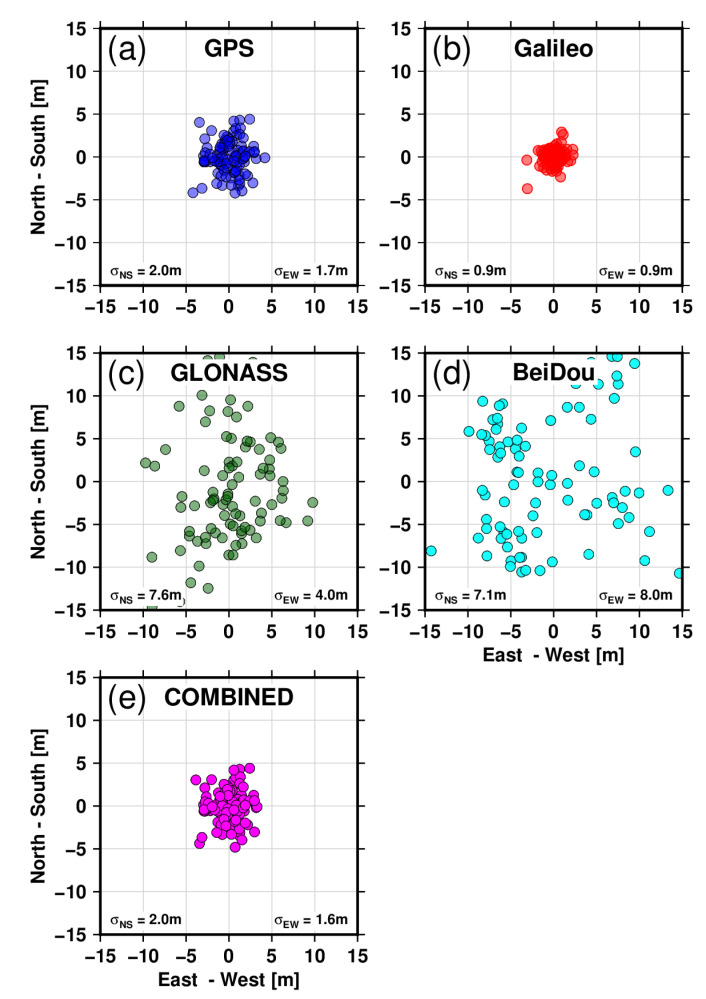
Single-point positioning for site KBG2, which has no Eccosorb materials mounted to its antenna again using different GNSS constellations. Results are shown in blue for GPS (**a**), in red for Galileo (**b**), in dark green for GLONASS (**c**) and cyan for BeiDou (**d**), and in the magenta for COMBINED (**e**). The positioning accuracy using GPS in the north-south direction is 2.0 m and in the east-west direction, it is 1.7 m; for Galileo, it is 0.9 m and 0.9 m respectively; for GLONASS, it is 7.6 m and 4.0 m; and the lowest accuracy comes again from BeiDou, with 7.1 m and 8.0 m. We have also derived a combined GPS + GLONASS + Galileo + BeiDou solution with a north-south direction accuracy of 2.0 m and an east-west direction accuracy of 1.6 m.

**Figure 13 sensors-22-03384-f013:**
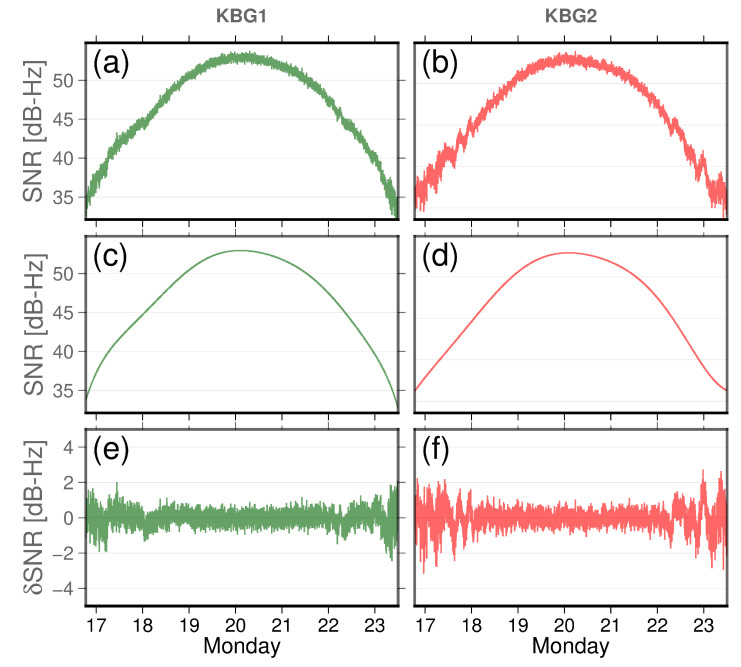
Raw SNR time series where the direct and indirect signals are superimposed as observed by the Alloy Trimble reference receiver for the L1, the GPS signal on 1 March, for GPS satellite PRN27. (**a**) Raw SNR as recorded by station KBG1. (**b**) Raw SNR as recorded by station KBG2, where the direct signal for both stations dominates the main signal. We have applied the ninth-order polynomial as depicted in (**c**) for KBG1 and (**d**) for KBG2. After subtracting the polynomial fit of the model from the raw SNR values, δSNR is shown in (**e**) for KBG1 and (**f**) for KBG2. Horizontal axes are the time of day in units of hours. The vertical axes are SNRs in units of dB-Hz.

**Figure 14 sensors-22-03384-f014:**
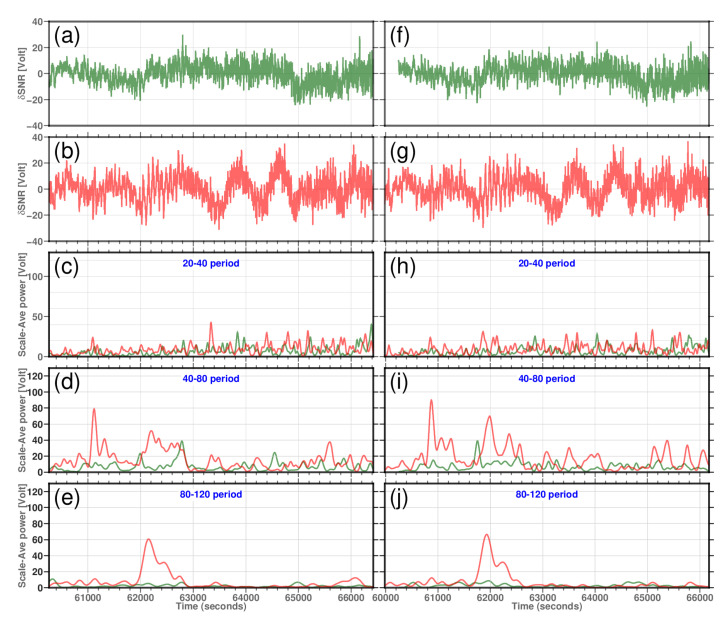
The Morlet wavelet analysis of δSNR for an ascending arc of GPS satellite PRN27 on 1 March 2021. (**a**,**b**) δSNR time series (solid light green line) for station KBG1 for days of the year (DoYs) 60 and 61, respectively. (**c**,**d**) δSNR time series (solid light red line) for station KBG2 for doYs 60 and 61, respectively. (**e**,**f**) Scaled averaged wavelet power $\left(\frac{|{\left(W_{n}\left(s\right)|\right)}^{2}}{s}\right)$ for all scales, which appears in the 20–40 s period band for stations KBG1 and KBG2 for DoYs 60 and 61, respectively. (**g**,**h**) Scaled averaged wavelet power for all scales, which appears for the period 40–80 s for both stations during DoYs 60 and 61, respectively. The maximum powers for station KBG2 appear at 61,118 s and 60,872 s for DoYs 60 and 61, respectively. A second peak with diminished power also appears for KBG2. For KBG1, the maximum powers appear at 62,786 and 61,754 s for the DoYs 60 and 61, respectively, with much-reduced power compared to KBG2. (**i**,**j**) Scaled averaged wavelet power for all scales, which appear in the 80–120 s band for both stations and for the DoYs 60 and 61, respectively. For KBG2, the maximum power occurs at 62,152 s for DoY 060 and 61,930 s for DoY 61.

**Figure 15 sensors-22-03384-f015:**
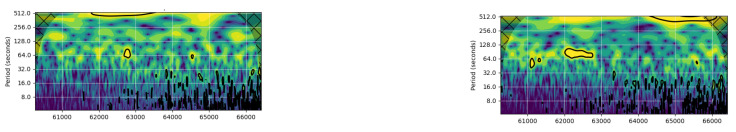
The wavelet power spectrum of residual SNR time series for station KBG1 (**left**) as depicted in Figure 14a with light green solid line and for station KBG2 (**right**) as depicted in Figure 14c with a light red solid line as observed for KBG1 and KBG2 on 1 March 2021 (DoY 60), as the satellite GPS PRN27 ascends, subtending the angles between 255 and 275${}^{\circ }$ in azimuth. The Morlet wavelet is used to create a wavelet spectrum. The left axis is the period in seconds that corresponds to wavelet scales with base 2. The horizontal axes represent time (seconds). The representations of the Morlet wavelet spectrum show a significant power for KBG2 for the periods 30–64 and 64–128 s, spanning approximately 100 and 850 s, respectively. However, for the station KBG1, only a short-lived power is observed for the period 60–100 s. The thick contour encloses areas of higher than 95% confidence. The cross-hatched portions represent the cone of influence, where the edge effects become substantial.

## Data Availability

The RINEX data is available from the authors upon reasonable request.

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
