# Peer review of "Evaluation of the Multipath Environment Using Electromagnetic-Absorbing Materials at Continuous GNSS Stations"

_sensors, 2022, doi:10.3390/s22093384_

Round 1

Reviewer 1 Report

 Minor remarks:

  1. The description of the chapter structure (row 134-140) in the introduction part does not match the manuscript content. The contribution of the chapter "Fresnel Zone" to the manuscript is unclear. Please check it.
  2. The GNSS stations "KBG1 and KBG2" (row 215) were mentioned before they were introduced.
  3. The post-fit residuals (row 265) and PPP processing results (row 288) of the two stations were not shown.
  4. If possible, authors can consider the effect of multi-system combined single point positioning in Section 6.1.
  5. “With the lowest accuracy coming from BeiDou at 40.9 m and 56.2 m” (row 426), please check or explain the reasons for the extremely low accuracy of the Beidou system.
  6. Authors should add more details to Figure 15.

Author Response

Thank you for the opportunity to revise our manuscript. The suggestions and comments offered by the reviewer are immensely helpful, and we also appreciate your insightful suggestions and comments on revising the aspects of the manuscript.

The description of the chapter structure (row 134-140) in the introduction part does not match the manuscript content. The contribution of the chapter "Fresnel Zone" to the manuscript is unclear. Please check it.

  The GNSS stations "KBG1 and KBG2" (row 215) were mentioned before they were introduced.

Reply: We appreciate the reviewer for pointing out this. We have now added a description of the station's name, and further introductions to the stations are also indicated.

 The post-fit residuals (row 265) and PPP processing results (row 288) of the two stations were not shown.

Reply: Corrections have been made. We have now incorporated a wording to the reader that the post-fit residuals are shown in the section specified. We have used two independent GNSS scientific software packages to study post-fit residuals, and the results are consistent between the two GNSS packages. But we have not shown the results from the JPL software package in the manuscript, and we have qualified this in section 6.

 If possible, authors can consider the effect of multi-system combined single point positioning in Section 6.1.

Reply: We think this is an excellent suggestion. We have now incorporated the solution of the single point positioning for the combined solution (multi-GNSS) and included it in the manuscript. These changes are reflected in Figures 11 and 12 of the manuscript, including in the text.

 “With the lowest accuracy coming from BeiDou at 40.9 m and 56.2 m” (row 426), please check or explain the reasons for the extremely low accuracy of the Beidou system.

Reply: Thank you for pointing this out. The standard errors to local coordinates (N-S and E-W) with respect to a reference coordinate indeed show the lowest accuracy for the Beidou system compared to the other constellations. The larger values indicated in the initial round of the manuscript has indeed error. The standard errors are now less than 10m, and these values are reflected in the revised manuscript (an error occurred due to the use of the newer version of RINEX 3.04 format). We alluded to the low accuracy of the Beidou standard point solutions to the quality of the broadcast orbit (e.g., GEO satellites), which may result from large code multipath errors for the BDS system.

 Authors should add more details to Figure 15.

Reply. Now we have added more details to the figure captions.

Reviewer 2 Report

Dear Authors,

the article entitled: Evaluation of the Multipath Environment Using Electromagnetic-absorbing Materials at Continuous GNSS Stations paper presents the use an Eccosorb AN-W-79 microwave absorbing material mounted around a Global Navigation Satellite System (GNSS) antenna that reflects less than -17 dB of normal incident energy above a frequency of 600 MHz. It is very important because to date, no universal modelling technique is available to mitigate the effect of site-specific multipaths in high-precision GNSS data processing.

All chapters (abstract, introduction, multipath theory and signal-to-noise ratio, fresnel zones, experimental setup and data acquisition, impacts of the microwave absorption: observation results, effects of multipath on GNSS carrier post-fit residuals, as well as conclusion) are very well described and they do not raise any doubts. In terms of the literature review is very sufficient (64 positions), all of which are papers from recognized scientific journals, such as: GPS Solutions, Journal of Geophysical Research, Radio Science, and others. Moreover, I would like to point out that the papers cited are related to the subject of this article (GNSS, multipath, site-specific effects, MAM and GNSS reference station). However, in the publication make the following changes:

  • I propose to extend the literature in the first sentence of the introduction, related to the applications of GNSS, such as for example:
  1. Gao, Z.; Ge, M.; Li, Y.; Shen, W.; Zhang, H.; Schuh, H. Railway Irregularity Measuring Using Rauch–Tung–Striebel Smoothed Multi-sensors Fusion System: Quad-GNSS PPP, IMU, Odometer, and Track Gauge. GPS Solut. 201822, 36.
  2. Liu, W.; Shi, X.; Zhu, F.; Tao, X.; Wang, F. Quality Analysis of Multi-GNSS Raw Observations and a Velocity-aided Positioning Approach Based on Smartphones. Adv. Space Res. 201963, 2358–2377.
  3. Specht, C.; Specht, M.; DÄ…browski, P. Comparative Analysis of Active Geodetic Networks in Poland. In Proceedings of the 17th International Multidisciplinary Scientific GeoConference (SGEM 2017), Albena, Bulgaria, 27 June–6 July 2017.
  • Please write sentences impersonally.

To sum up, after taking into account the above amendments (minor revision), I suppose that this article is suitable for publication in the Sensors.

Author Response

We highly appreciate the reviewer’s helpful comments to revise our manuscript.

1.  Gao, Z.; Ge, M.; Li, Y.; Shen, W.; Zhang, H.; Schuh, H. Railway Irregularity Measuring Using Rauch–Tung–Striebel Smoothed Multi-sensors Fusion System: Quad-GNSS PPP, IMU, Odometer, and Track Gauge. GPS Solut. 201822, 36.

2.  Liu, W.; Shi, X.; Zhu, F.; Tao, X.; Wang, F. Quality Analysis of Multi-GNSS Raw Observations and a Velocity-aided Positioning Approach Based on Smartphones. Adv. Space Res. 201963, 2358–2377.

3. Specht, C.; Specht, M.; DÄ…browski, P. Comparative Analysis of Active Geodetic Networks in Poland. In Proceedings of the 17th International Multidisciplinary Scientific GeoConference (SGEM 2017), Albena, Bulgaria, 27 June–6 July 2017.

Reply: Thank you for pointing this out. We have included all the recommended references as well as additional references to the manuscript.

Please write sentences impersonally.

Reply: Some of the sentences have been recast in a more impersonal manner.  A crucial point has been mentioned by the reviewer. However, some of the sentences have been maintained because the adjustments make the sentences passive. In general, we feel that sentences should be written in the active voice.

To sum up, after taking into account the above amendments (minor revision), I suppose that this article is suitable for publication in the Sensors.

We thank you for your valuable comments.

Sincerely yours,

Addisu and Norman

Reviewer 3 Report

This work  is an interesting work and presents valuable results on  Multipath Environment effect on GNSS Stations. The subject of the work is very important for a better understanding of the processes of site-specific multipaths in high-precision GNSS data processing. I recommand to publish it with sligh modify.

  1. In someplace (e.g. mainly due to the significant advancement in high-accuracy positioning products.) need some literatures, 
    e.g.: Review papers on GNSS Physical applications of GPS geodesy: A review-2016; Review of current GPS methodologies for producing accurate time series and their error sources-2017.
  2. The references need in order, e.g.: multipath sources [4,18], the order of the citation is in disorder. we did not see [1], it starts at [4].
  3. the author need check the whole chapter carefully.

Author Response

This work is an interesting work and presents valuable results on the Multipath Environment effect on GNSS Stations. The subject of the work is very important for a better understanding of the processes of site-specific multipaths in high-precision GNSS data processing. I recommand to publish it with sligh modify.

Thank you for the opportunity to revise our manuscript. The suggestions and comments offered by the reviewer are very helpful, and we also appreciate your insightful suggestions and comments on revising the aspects of the manuscript.

 1. In someplace (e.g. mainly due to the significant advancement in high-accuracy positioning products.) need some literatures, e.g.: Review papers on GNSS Physical applications of GPS geodesy: A review-2016; Review of current GPS methodologies for producing accurate time series and their error sources-2017.

Reply: We appreciate the reviewer for pointing out this. We have added a few appropriate references, particularly at the beginning of the introduction section.

2. The references need in order, e.g.: multipath sources [4,18], the order of the citation is in disorder. we did not see [1], it starts at [4].

Reply:  We again appreciate the reviewer for pointing this out. We have now corrected this.

3. The author need check the whole chapter carefully.

Reply. We agree with this and we have checked throughout the manuscript. We believe that the ordering has been done correctly.

Sincerely yours,

Addisu and Norman